# THE REACTOR:
# A FAST AND SAMPLE-EFFICIENT ACTOR-CRITIC AGENT FOR REINFORCEMENT LEARNING

**Audrūnas Gruslys,**
DeepMind
audrunas@google.com

**Will Dabney,**
DeepMind
wdabney@google.com

**Mohammad Gheshlaghi Azar,**
DeepMind
mazar@google.com

**Bilal Piot,**
DeepMind
piot@google.com

**Marc G. Bellemare,**
Google Brain
bellemare@google.com

**Rémi Munos,**
DeepMind
munos@google.com

## ABSTRACT

In this work, we present a new agent architecture, called Reactor, which combines multiple algorithmic and architectural contributions to produce an agent with higher sample-efficiency than Prioritized Dueling DQN (Wang et al., 2017) and Categorical DQN (Bellemare et al., 2017), while giving better run-time performance than A3C (Mnih et al., 2016). Our first contribution is a new policy evaluation algorithm called Distributional Retrace, which brings multi-step off-policy updates to the distributional reinforcement learning setting. The same approach can be used to convert several classes of multi-step policy evaluation algorithms, designed for expected value evaluation, into distributional algorithms. Next, we introduce the *β-leave-one-out* policy gradient algorithm, which improves the trade-off between variance and bias by using action values as a baseline. Our final algorithmic contribution is a new prioritized replay algorithm for sequences, which exploits the temporal locality of neighboring observations for more efficient replay prioritization. Using the Atari 2600 benchmarks, we show that each of these innovations contribute to both sample efficiency and final agent performance. Finally, we demonstrate that Reactor reaches state-of-the-art performance after 200 million frames and less than a day of training.

## 1 INTRODUCTION

Model-free deep reinforcement learning has achieved several remarkable successes in domains ranging from super-human-level control in video games (Mnih et al., 2015) and the game of Go (Silver et al., 2016; 2017), to continuous motor control tasks (Lillicrap et al., 2015; Schulman et al., 2015).

Much of the recent work can be divided into two categories. First, those of which that, often building on the DQN framework, act $\epsilon$-greedily according to an action-value function and train using mini-batches of transitions sampled from an experience replay buffer (Van Hasselt et al., 2016; Wang et al., 2015; He et al., 2017; Anschel et al., 2017). These *value-function agents* benefit from improved sample complexity, but tend to suffer from long runtimes (e.g. DQN requires approximately a week to train on Atari). The second category are the *actor-critic agents*, which includes the asynchronous advantage actor-critic (A3C) algorithm, introduced by Mnih et al. (2016). These agents train on transitions collected by multiple actors running, and often training, in parallel (Schulman et al., 2017; Vezhnevets et al., 2017). The deep actor-critic agents train on each trajectory only once, and thus tend to have worse sample complexity. However, their distributed nature allows significantly faster training in terms of wall-clock time. Still, not all existing algorithms can be put in the above two categories and various hybrid approaches do exist (Zhao et al., 2016; O'Donoghue et al., 2017; Gu et al., 2017; Wang et al., 2017).

Data-efficiency and off-policy learning are essential for many real-world domains where interactions with the environment are expensive. Similarly, wall-clock time (time-efficiency) directly impacts an algorithm's applicability through resource costs. The focus of this work is to produce an agent that is sample- and time-efficient. To this end, we introduce a new reinforcement learning agent, called *Reactor* (Retrace-Actor), which takes a principled approach to combining the sample-efficiency of off-policy experience replay with the time-efficiency of asynchronous algorithms. We combine recent advances in both categories of agents with novel contributions to produce an agent that inherits the benefits of both and reaches state-of-the-art performance over 57 Atari 2600 games.

Our primary contributions are (1) a novel policy gradient algorithm, $\beta$-LOO, which makes better use of action-value estimates to improve the policy gradient; (2) the first multi-step off-policy distributional reinforcement learning algorithm, distributional Retrace($\lambda$); (3) a novel prioritized replay for off-policy sequences of transitions; and (4) an optimized network and parallel training architecture.

We begin by reviewing background material, including relevant improvements to both value-function agents and actor-critic agents. In Section 3 we introduce each of our primary contributions and present the Reactor agent. Finally, in Section 4, we present experimental results on the 57 Atari 2600 games from the Arcade Learning Environment (ALE) (Bellemare et al., 2013), as well as a series of ablation studies for the various components of Reactor.

## 2 BACKGROUND

We consider a Markov decision process (MDP) with state space $X$ and finite action space $\mathcal{A}$. A (stochastic) policy $\pi(\cdot|x)$ is a mapping from states $x \in X$ to a probability distribution over actions. We consider a $\gamma$-discounted infinite-horizon criterion, with $\gamma \in [0, 1)$ the discount factor, and define for policy $\pi$ the action-value of a state-action pair $(x, a)$ as

$$Q^\pi(x, a) \stackrel{\text{def}}{=} \mathbb{E}\Big[\sum_{t \geq 0} \gamma^t r_t | x_0 = x, a_0 = a, \pi\Big],$$

where $(\{x_t\}_{t \geq 0})$ is a trajectory generated by choosing $a$ in $x$ and following $\pi$ thereafter, i.e., $a_t \sim \pi(\cdot|x_t)$ (for $t \geq 1$), and $r_t$ is the reward signal. The objective in reinforcement learning is to find an optimal policy $\pi^*$, which maximises $Q^\pi(x, a)$. The optimal action-values are given by $Q^*(x, a) = \max_\pi Q^\pi(x, a)$.

### 2.1 VALUE-BASED ALGORITHMS

The Deep Q-Network (DQN) framework, introduced by Mnih et al. (2015), popularised the current line of research into deep reinforcement learning by reaching human-level, and beyond, performance across 57 Atari 2600 games in the ALE. While DQN includes many specific components, the essence of the framework, much of which is shared by Neural Fitted Q-Learning (Riedmiller, 2005), is to use of a deep convolutional neural network to approximate an action-value function, training this approximate action-value function using the Q-Learning algorithm (Watkins & Dayan, 1992) and mini-batches of one-step transitions $(x_t, a_t, r_t, x_{t+1}, \gamma_t)$ drawn randomly from an experience replay buffer (Lin, 1992). Additionally, the next-state action-values are taken from a *target network*, which is updated to match the current network periodically. Thus, the temporal difference (TD) error for transition $t$ used by these algorithms is given by

$$\delta_t = r_t + \gamma_t \max_{a' \in \mathcal{A}} Q(x_{t+1}, a'; \bar{\theta}) - Q(x_t, a_t; \theta), \tag{1}$$

where $\theta$ denotes the parameters of the network and $\bar{\theta}$ are the parameters of the target network.

Since this seminal work, we have seen numerous extensions and improvements that all share the same underlying framework. Double DQN (Van Hasselt et al., 2016), attempts to correct for the over-estimation bias inherent in Q-Learning by changing the second term of (1) to $Q(x_{t+1}, \arg\max_{a' \in \mathcal{A}} Q(x_{t+1}, a'; \theta); \bar{\theta})$. The dueling architecture (Wang et al., 2015), changes the network to estimate action-values using separate network heads $V(x; \theta)$ and $A(x, a; \theta)$ with

$$Q(x, a; \theta) = V(x; \theta) + A(x, a; \theta) - \frac{1}{|A|} \sum_{a'} A(x, a'; \theta).$$

Recently, Hessel et al. (2017) introduced Rainbow, a value-based reinforcement learning agent combining many of these improvements into a single agent and demonstrating that they are largely complementary. Rainbow significantly out performs previous methods, but also inherits the poorer time-efficiency of the DQN framework. We include a detailed comparison between Reactor and Rainbow in the Appendix. In the remainder of the section we will describe in more depth other recent improvements to DQN.

### 2.1.1 PRIORITIZED EXPERIENCE REPLAY

The experience replay buffer was first introduced by Lin (1992) and later used in DQN (Mnih et al., 2015). Typically, the replay buffer is essentially a first-in-first-out queue with new transitions gradually replacing older transitions. The agent would then sample a mini-batch uniformly at random from the replay buffer. Drawing inspiration from prioritized sweeping (Moore & Atkeson, 1993), prioritized experience replay replaces the uniform sampling with prioritized sampling proportional to the absolute TD error (Schaul et al., 2016).

Specifically, for a replay buffer of size $N$, prioritized experience replay samples transition $t$ with probability $P(t)$, and applies weighted importance-sampling with $w_t$ to correct for the prioritization bias, where

$$P(t) = \frac{p_t^\alpha}{\sum_k p_k^\alpha}, \quad w_t = \left(\frac{1}{N} \cdot \frac{1}{P(t)}\right)^\beta, \quad p_t = |\delta_t| + \epsilon, \quad \alpha, \beta, \epsilon > 0. \tag{2}$$

Prioritized DQN significantly increases both the sample-efficiency and final performance over DQN on the Atari 2600 benchmarks (Schaul et al., 2015).

### 2.1.2 RETRACE($\lambda$)

Retrace($\lambda$) is a convergent off-policy multi-step algorithm extending the DQN agent (Munos et al., 2016). Assume that some trajectory $\{x_0, a_0, r_0, x_1, a_1, r_1, \ldots, x_t, a_t, r_t, \ldots, \}$ has been generated according to *behaviour policy* $\mu$, i.e., $a_t \sim \mu(\cdot|x_t)$. Now, we aim to evaluate the value of a different *target policy* $\pi$, i.e. we want to estimate $Q^\pi$. The Retrace algorithm will update our current estimate $Q$ of $Q^\pi$ in the direction of

$$\Delta Q(x_t, a_t) \overset{\text{def}}{=} \sum_{s \geq t} \gamma^{s-t}(c_{t+1} \ldots c_s)\delta_s^\pi Q, \tag{3}$$

where $\delta_s^\pi Q \overset{\text{def}}{=} r_s + \gamma \mathbb{E}_\pi[Q(x_{s+1}, \cdot)] - Q(x_s, a_s)$ is the temporal difference at time $s$ under $\pi$, and

$$c_s = \lambda \min(1, \rho_s), \quad \rho_s = \frac{\pi(a_s|x_s)}{\mu(a_s|x_s)}. \tag{4}$$

The Retrace algorithm comes with the theoretical guarantee that in finite state and action spaces, repeatedly updating our current estimate $Q$ according to (3) produces a sequence of Q functions which converges to $Q^\pi$ for a fixed $\pi$ or to $Q^*$ if we consider a sequence of policies $\pi$ which become increasingly greedy w.r.t. the $Q$ estimates (Munos et al., 2016).

### 2.1.3 DISTRIBUTIONAL RL

Distributional reinforcement learning refers to a class of algorithms that directly estimate the distribution over returns, whose expectation gives the traditional value function (Bellemare et al., 2017). Such approaches can be made tractable with a distributional Bellman equation, and the recently proposed algorithm $C51$ showed state-of-the-art performance in the Atari 2600 benchmarks. $C51$ parameterizes the distribution over returns with a mixture over Diracs centered on a uniform grid,

$$Q(x, a; \theta) = \sum_{i=0}^{N-1} q_i(x, a; \theta)z_i, \quad q_i = \frac{e^{\theta_i(x,a)}}{\sum_{j=0}^{N-1} e^{\theta_j(x,a)}}, \quad z_i = v_{\min} + i \frac{v_{\max} - v_{\min}}{N-1}, \tag{5}$$

with hyperparameters $v_{\min}, v_{\max}$ that bound the distribution support of size $N$.

## 2.2 ACTOR-CRITIC ALGORITHMS

In this section we review the actor-critic framework for reinforcement learning algorithms and then discuss recent advances in actor-critic algorithms along with their various trade-offs. The asynchronous advantage actor-critic (A3C) algorithm (Mnih et al., 2016), maintains a parameterized policy $\pi(a|x; \theta)$ and value function $V(x; \theta_v)$, which are updated with

$$\triangle\theta = \nabla_\theta \log \pi(a_t|x_t; \theta) A(x_t, a_t; \theta_v), \quad \triangle\theta_v = A(x_t, a_t; \theta_v) \nabla_{\theta_v} V(x_t), \quad (6)$$

$$\text{where,} \quad A(x_t, a_t; \theta_v) = \sum_k^{n-1} \gamma^k r_{t+k} + \gamma^n V(x_{t+n}) - V(x_t). \quad (7)$$

A3C uses $M = 16$ parallel CPU workers, each acting independently in the environment and applying the above updates asynchronously to a shared set of parameters. In contrast to the previously discussed value-based methods, A3C is an on-policy algorithm, and does not use a GPU nor a replay buffer.

Proximal Policy Optimization (PPO) is a closely related actor-critic algorithm (Schulman et al., 2017), which replaces the advantage (7) with,

$$\min(\rho_t A(x_t, a_t; \theta_v), clip(\rho_t, 1 - \epsilon, 1 + \epsilon) A(x_t, a_t; \theta_v)), \ \epsilon > 0,$$

where $\rho_t$ is as defined in Section 2.1.2. Although both PPO and A3C run $M$ parallel workers collecting trajectories independently in the environment, PPO collects these experiences to perform a single, synchronous, update in contrast with the asynchronous updates of A3C.

Actor-Critic Experience Replay (ACER) extends the A3C framework with an experience replay buffer, Retrace algorithm for off-policy corrections, and the Truncated Importance Sampling Likelihood Ratio (TISLR) algorithm used for off-policy policy optimization (Wang et al., 2017).

## 3 THE REACTOR

The Reactor is a combination of four novel contributions on top of recent improvements to both deep value-based RL and policy-gradient algorithms. Each contribution moves Reactor towards our goal of achieving both sample and time efficiency.

### 3.1 $\beta$-LOO

The Reactor architecture represents both a policy $\pi(a|x)$ and action-value function $Q(x, a)$. We use a policy gradient algorithm to train the actor $\pi$ which makes use of our current estimate $Q(x, a)$ of $Q^\pi(x, a)$. Let $V^\pi(x_0)$ be the value function at some initial state $x_0$, the policy gradient theorem says that $\nabla V^\pi(x_0) = \mathbb{E}\big[\sum_t \gamma^t \sum_a Q^\pi(x_t, a)\nabla\pi(a|x_t)\big]$, where $\nabla$ refers to the gradient w.r.t. policy parameters (Sutton et al., 2000). We now consider several possible ways to estimate this gradient.

To simplify notation, we drop the dependence on the state $x$ for now and consider the problem of estimating the quantity

$$G = \sum_a Q^\pi(a)\nabla\pi(a). \quad (8)$$

In the off-policy case, we consider estimating $G$ using a single action $\hat{a}$ drawn from a (possibly different from $\pi$) behaviour distribution $\hat{a} \sim \mu$. Let us assume that for the chosen action $\hat{a}$ we have access to an unbiased estimate $R(\hat{a})$ of $Q^\pi(\hat{a})$. Then, we can use likelihood ratio (LR) method combined with an importance sampling (IS) ratio (which we call ISLR) to build an unbiased estimate of $G$:

$$\hat{G}_{\text{ISLR}} = \frac{\pi(\hat{a})}{\mu(\hat{a})}(R(\hat{a}) - V)\nabla\log\pi(\hat{a}),$$

where $V$ is a baseline that depends on the state but not on the chosen action. However this estimate suffers from high variance. A possible way for reducing variance is to estimate $G$ directly from (8) by using the return $R(\hat{a})$ for the chosen action $\hat{a}$ and our current estimate $Q$ of $Q^\pi$ for the other actions, which leads to the so-called *leave-one-out* (LOO) policy-gradient estimate:

$$\hat{G}_{\text{LOO}} = R(\hat{a})\nabla\pi(\hat{a}) + \sum_{a \neq \hat{a}} Q(a)\nabla\pi(a). \quad (9)$$

Figure 1: Single-step (left) and multi-step (right) distribution bootstrapping.

This estimate has low variance but may be biased if the estimated $Q$ values differ from $Q^\pi$. A better bias-variance tradeoff may be obtained by the more general $\beta$-*LOO policy-gradient* estimate:

$$\hat{G}_{\beta\text{-LOO}} = \beta(R(\hat{a}) - Q(\hat{a}))\nabla\pi(\hat{a}) + \sum_a Q(a)\nabla\pi(a), \qquad (10)$$

where $\beta = \beta(\mu, \pi, \hat{a})$ can be a function of both policies, $\pi$ and $\mu$, and the selected action $\hat{a}$. Notice that when $\beta = 1$, (10) reduces to (9), and when $\beta = 1/\mu(\hat{a})$, then (10) is

$$\hat{G}_{\frac{1}{\mu}\text{-LOO}} = \frac{\pi(\hat{a})}{\mu(\hat{a})}(R(\hat{a}) - Q(\hat{a}))\nabla\log\pi(\hat{a}) + \sum_a Q(a)\nabla\pi(a). \qquad (11)$$

This estimate is unbiased and can be seen as a generalization of $\hat{G}_{\text{ISLR}}$ where instead of using a state-only dependent baseline, we use a state-and-action-dependent baseline (our current estimate $Q$) and add the correction term $\sum_a \nabla\pi(a)Q(a)$ to cancel the bias. Proposition 1 gives our analysis of the bias of $G_{\beta\text{-LOO}}$, with a proof left to the Appendix.

**Proposition 1.** *Assume $\hat{a} \sim \mu$ and that $\mathbb{E}[R(\hat{a})] = Q^\pi(\hat{a})$. Then, the bias of $G_{\beta\text{-LOO}}$ is $\left|\sum_a(1 - \mu(a)\beta(a))\nabla\pi(a)[Q(a) - Q^\pi(a)]\right|$.*

Thus the bias is small when $\beta(a)$ is close to $1/\mu(a)$, or when the $Q$-estimates are close to the true $Q^\pi$ values, and unbiased regardless of the estimates if $\beta(a) = 1/\mu(a)$. The variance is low when $\beta$ is small, therefore, in order to improve the bias-variance tradeoff we recommend using the $\beta$-LOO estimate with $\beta$ defined as: $\beta(\hat{a}) = \min\left(c, \frac{1}{\mu(\hat{a})}\right)$, for some constant $c \geq 1$. This truncated $1/\mu$ coefficient shares similarities with the truncated IS gradient estimate introduced in (Wang et al., 2017) (which we call TISLR for truncated-ISLR):

$$\hat{G}_{\text{TISLR}} = \min\left(c, \frac{\pi(\hat{a})}{\mu(\hat{a})}\right)(R(\hat{a}) - V)\nabla\log\pi(\hat{a}) + \sum_a \left(\frac{\pi(a)}{\mu(a)} - c\right)_+ \mu(a)(Q^\pi(a) - V)\nabla\log\pi(a).$$

The differences are: (i) we truncate $1/\mu(\hat{a}) = \pi(\hat{a})/\mu(\hat{a}) \times 1/\pi(\hat{a})$ instead of truncating $\pi(\hat{a})/\mu(\hat{a})$, which provides an additional variance reduction due to the variance of the LR $\nabla\log\pi(\hat{a}) = \frac{\nabla\pi(\hat{a})}{\pi(\hat{a})}$ (since this LR may be large when a low probability action is chosen), and (ii) we use our $Q$-baseline instead of a $V$ baseline, reducing further the variance of the LR estimate.

## 3.2 DISTRIBUTIONAL RETRACE

In off-policy learning it is very difficult to produce an unbiased sample $R(\hat{a})$ of $Q^\pi(\hat{a})$ when following another policy $\mu$. This would require using full importance sampling correction along the trajectory. Instead, we use the off-policy corrected return computed by the Retrace algorithm, which produces a (biased) estimate of $Q^\pi(\hat{a})$ but whose bias vanishes asymptotically (Munos et al., 2016).

In Reactor, we consider predicting an approximation of the return distribution function from any state-action pair $(x, a)$ in a similar way as in Bellemare et al. (2017). The original algorithm C51 described in that paper considered single-step Bellman updates only. Here we need to extend this idea to multi-step updates and handle the off-policy correction performed by the Retrace algorithm, as defined in (3). Next, we describe these two extensions.

**Multi-step distributional Bellman operator:** First, we extend C51 to multi-step Bellman backups. We consider return-distributions from $(x, a)$ of the form $\sum_i q_i(x, a)\delta_{z_i}$ (where $\delta_z$ denotes a Dirac in $z$)

which are supported on a finite uniform grid $\{z_i\} \in [v_{\min}, v_{\max}]$, $z_i < z_{i+1}$, $z_1 = v_{\min}$, $z_m = v_{\max}$. The coefficients $q_i(x, a)$ (discrete distribution) corresponds to the probabilities assigned to each atom $z_i$ of the grid. From an observed $n$-step sequence $\{x_t, a_t, r_t, x_{t+1}, \ldots, x_{t+n}\}$, generated by behavior policy $\mu$ (i.e, $a_s \sim \mu(\cdot|x_s)$ for $t \leq s < t + n$), we build the $n$-step backed-up return-distribution from $(x_t, a_t)$. The $n$-step distributional Bellman target, whose expectation is $\sum_{s=t}^{t+n-1} \gamma^{s-t} r_s + \gamma^n Q(x_{t+n}, a)$, is given by:

$$\sum_i q_i(x_{t+n}, a)\delta_{z_i^n}, \text{ with } z_i^n = \sum_{s=t}^{t+n-1} \gamma^{s-t} r_s + \gamma^n z_i.$$

Since this distribution is supported on the set of atoms $\{z_i^n\}$, which is not necessarily aligned with the grid $\{z_i\}$, we do a projection step and minimize the KL-loss between the projected target and the current estimate, just as with C51 except with a different target distribution (Bellemare et al., 2017).

**Distributional Retrace:** Now, the Retrace algorithm defined in (3) involves an off-policy correction which is not handled by the previous $n$-step distributional Bellman backup. The key to extending this distributional back-up to off-policy learning is to rewrite the Retrace algorithm as a linear combination of $n$-step Bellman backups, weighted by some coefficients $\alpha_{n,a}$. Indeed, notice that (3) rewrites as

$$\Delta Q(x_t, a_t) = \sum_{n \geq 1} \sum_{a \in \mathcal{A}} \alpha_{n,a} \underbrace{\Big[ \sum_{s=t}^{t+n-1} \gamma^{s-t} r_s + \gamma^n Q(x_{t+n}, a) \Big]}_{n\text{-step Bellman backup}} - Q(x_t, a_t),$$

where $\alpha_{n,a} = \big(c_{t+1} \ldots c_{t+n-1}\big)\big(\pi(a|x_{t+n}) - \mathbb{I}\{a = a_{t+n}\}c_{t+n}\big)$. These coefficients depend on the degree of off-policy-ness (between $\mu$ and $\pi$) along the trajectory. We have that $\sum_{n \geq 1} \sum_a \alpha_{n,a} = \sum_{n \geq 1}\big(c_{t+1} \ldots c_{t+n-1}\big)(1 - c_{t+n}) = 1$, but notice some coefficients may be negative. However, in expectation (over the behavior policy) they are non-negative. Indeed,

$$\begin{aligned}
\mathbb{E}_\mu[\alpha_{n,a}] &= \mathbb{E}\Big[ \big(c_{t+1} \ldots c_{t+n-1}\big)\mathbb{E}_{a_{t+n} \sim \mu(\cdot|x_{t+n})}\big[\pi(a|x_{t+n}) - \mathbb{I}\{a = a_{t+n}\}c_{t+n}|x_{t+n}\big]\Big] \\
&= \mathbb{E}\Big[ \big(c_{t+1} \ldots c_{t+n-1}\big)\Big(\pi(a|x_{t+n}) - \mu(a|x_{t+n})\lambda \min\big(1, \tfrac{\pi(a|x_{t+n})}{\mu(a|x_{t+n})}\big)\Big)\Big] \geq 0,
\end{aligned}$$

by definition of the $c_s$ coefficients (4). Thus in expectation (over the behavior policy), the Retrace update can be seen as a *convex* combination of $n$-step Bellman updates.

Then, the distributional Retrace algorithm can be defined as backing up a *mixture* of $n$-step distributions. More precisely, we define the Retrace target distribution as:

$$\sum_{i=1} q_i^*(x_t, a_t)\delta_{z_i}, \text{ with } q_i^*(x_t, a_t) = \sum_{n \geq 1} \sum_a \alpha_{n,a} \sum_j q_j(x_{t+n}, a_{t+n})h_{z_i}(z_j^n),$$

where $h_{z_i}(x)$ is a linear interpolation kernel, projecting onto the support $\{z_i\}$:

$$h_{z_i}(x) = \left\{ \begin{array}{ll}
(x - z_{i-1})/(z_i - z_{i-1}), & \text{if } z_{i-1} \leq x \leq z_i \\
(z_{i+1} - x)/(z_{i+1} - z_i), & \text{if } z_i \leq x \leq z_{i+1} \\
0, & \text{if } x \leq z_{i-1} \text{ or } x \geq z_{i+1} \\
1, & \text{if } (x \leq v_{\min} \text{ and } z_i = v_{\min}) \text{ or } (x \geq v_{\max} \text{ and } z_i = v_{\max})
\end{array} \right\}$$

We update the current probabilities $q(x_t, a_t)$ by performing a gradient step on the KL-loss

$$\nabla \text{KL}(q^*(x_t, a_t), q(x_t, a_t)) = -\sum_{i=1} q_i^*(x_t, a_t)\nabla \log q_i(x_t, a_t). \tag{12}$$

Again, notice that some target "probabilities" $q_i^*(x_t, a_t)$ may be negative for some sample trajectory, but in expectation they will be non-negative. Since the gradient of a KL-loss is linear w.r.t. its first argument, our update rule (12) provides an unbiased estimate of the gradient of the KL between the expected (over the behavior policy) Retrace target distribution and the current predicted distribution.[1]

---

[1] We store past action probabilities $\mu$ together with actions taken in the replay memory.

**Remark:** The same method can be applied to other algorithms (such as TB($\lambda$) (Precup et al., 2000) and importance sampling (Precup et al., 2001)) in order to derive distributional versions of other off-policy multi-step RL algorithms.

### 3.3 PRIORITIZED SEQUENCE REPLAY

Prioritized experience replay has been shown to boost both statistical efficiency and final performance of deep RL agents (Schaul et al., 2016). However, as originally defined prioritized replay does not handle sequences of transitions and weights all unsampled transitions identically. In this section we present an alternative initialization strategy, called *lazy initialization*, and argue that it better encodes prior information about temporal difference errors. We then briefly describe our computationally efficient prioritized sequence sampling algorithm, with full details left to the appendix.

It is widely recognized that TD errors tend to be temporally correlated, indeed the need to break this temporal correlation has been one of the primary justifications for the use of experience replay (Mnih et al., 2015). Our proposed algorithm begins with this fundamental assumption.

**Assumption 1.** *Temporal differences are temporally correlated, with correlation decaying on average with the time-difference between two transitions.*

Prioritized experience replay adds new transitions to the replay buffer with a constant priority, but given the above assumption we can devise a better method. Specifically, we propose to add experience to the buffer with *no priority*, inserting a priority only after the transition has been sampled and used for training. Also, instead of sampling transitions, we assign priorities to all (overlapping) sequences of length $n$. When sampling, sequences with an assigned priority are sampled proportionally to that priority. Sequences with no assigned priority are sampled proportionally to the average priority of assigned priority sequences within some local neighbourhood. Averages are weighted to compensate for sampling biases (i.e. more samples are made in areas of high estimated priorities, and in the absence of weighting this would lead to overestimation of unassigned priorities).

The *lazy initialization* scheme starts with priorities $p_t$ corresponding to the sequences $\{x_t, a_t, \ldots, x_{t+n}\}$ for which a priority was already assigned. Then it extrapolates a priority of all other sequences in the following way. Let us define a partition $(I_i)_i$ of the states ordered by increasing time such that each cell $I_i$ contains exactly one state $s_i$ with already assigned priority. We define the estimated priority $\hat{p}_t$ to all other sequences as $\hat{p}_t = \sum_{s_i \in J(t)} \frac{w_i}{\sum_{i' \in J(t)} w_{i'}} p(s_i)$, where $J(t)$ is a collection of contiguous cells $(I_i)$ containing time $t$, and $w_i = |I_i|$ is the length of the cell $I_i$ containing $s_i$. For already defined priorities denote $\hat{p}_t = p_t$. Cell sizes work as estimates of inverse local density and are used as importance weights for priority estimation. [2] For the algorithm to be unbiased, partition $(I_i)_i$ must **not** be a function of the assigned priorities. So far we have defined a class of algorithms all free to choose the partition $(I_i)$ and the collection of cells $I(t)$, as long that they satisfy the above constraints. Figure 4 in the Appendix illustrates the above description.

Now, with probability $\epsilon$ we sample uniformly at random, and with probability $1 - \epsilon$ we sample proportionally to $\hat{p}_t$. We implemented an algorithm satisfying the above constraints and called it *Contextual Priority Tree* (CPT). It is based on AVL trees (Velskii & Landis, 1976) and can execute sampling, insertion, deletion and density evaluation in $O(\ln(n))$ time. We describe CPT in detail in the Appendix in Section 6.3.

We treated prioritization as purely a variance reduction technique. Importance-sampling weights were evaluated as in prioritized experience replay, with fixed $\beta = 1$ in (2). We used simple gradient magnitude estimates as priorities, corresponding to a mean absolute TD error along a sequence for Retrace, as defined in (3) for the classical RL case, and total variation in the distributional Retrace case.[3]

### 3.4 AGENT ARCHITECTURE

In order to improve CPU utilization we decoupled acting from learning. This is an important aspect of our architecture: an *acting thread* receives observations, submits actions to the environment, and

---

[2]Not to be confused with importance weights of produced samples.

[3]Sum of absolute discrete probability differences.

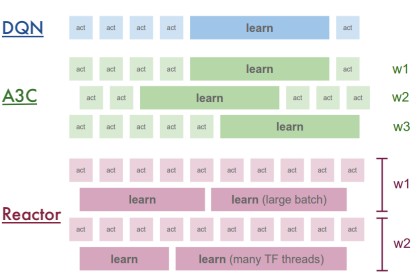

| Algorithm | Training Time | Type | # Workers |
|---|---|---|---|
| DQN | 8 days | GPU | 1 |
| Double DQN | 8 days | GPU | 1 |
| Dueling | 8 days | GPU | 1 |
| Prioritized DQN | 8 days | GPU | 1 |
| Rainbow | 10 days | GPU | 1 |
| A3C | 4 days | CPU | 16 |
| Reactor | **< 2 days** | CPU | 10+1 |
| Reactor 500m | 4 days | CPU | 10+1 |
| Reactor* | **< 1 day** | CPU | 20+1 |

Figure 2: (Left) The model of parallelism of DQN, A3C and Reactor architectures. Each row represents a separate thread. In Reactor's case, each worker, consiting of a learner and an actor is run on a separate worker machine. (Right) Comparison of training times and resources for various algorithms. 500m denotes 500 million training frames; otherwise 200m training frames were used.

stores transitions in memory, while a *learning thread* re-samples sequences of experiences from memory and trains on them (Figure 2, left). We typically execute 4-6 acting steps per each learning step. We sample sequences of length $n = 33$ in batches of 4. A moving network is unrolled over frames 1-32 while the target network is unrolled over frames 2-33.

We allow the agent to be distributed over multiple machines each containing action-learner pairs. Each worker downloads the newest network parameters before each learning step and sends delta-updates at the end of it. Both the network and target network are stored on a shared parameter server while each machine contains its own local replay memory. Training is done by downloading a shared network, evaluating local gradients and sending them to be applied on the shared network. While the agent can also be trained on a single machine, in this work we present results of training obtained with either 10 or 20 actor-learner workers and one parameter server. In Figure 2 (right) we compare resources and runtimes of Reactor with related algorithms.[4]

### 3.4.1 NETWORK ARCHITECTURE

In some domains, such as Atari, it is useful to base decisions on a short history of past observations. The two techniques generally used to achieve this are frame stacking and recurrent network architectures. We chose the latter over the former for reasons of implementation simplicity and computational efficiency. As the Retrace algorithm requires evaluating action-values over contiguous sequences of trajectories, using a recurrent architecture allowed each frame to be processed by the convolutional network only once, as opposed to $n$ times times if $n$ frame concatenations were used.

The Reactor architecture uses a recurrent neural network which takes an observation $x_t$ as input and produces two outputs: categorical action-value distributions $q_i(x_t, a)$ ($i$ here is a bin identifier), and policy probabilities $\pi(a|x_t)$. We use an architecture inspired by the duelling network architecture (Wang et al., 2015). We split action-value -distribution logits into state-value logits and advantage logits, which in turn are connected to the same LSTM network (Hochreiter & Schmidhuber, 1997). Final action-value logits are produced by summing state- and action-specific logits, as in Wang et al. (2015). Finally, a softmax layer on top for each action produces the distributions over discounted future returns.

The policy head uses a softmax layer mixed with a fixed uniform distribution over actions, where this mixing ratio is a hyperparameter (Wiering, 1999, Section 5.1.3). Policy and Q-networks have separate LSTMs. Both LSTMs are connected to a shared linear layer which is connected to a shared convolutional neural network (Krizhevsky et al., 2012). The precise network specification is given in Table 3 in the Appendix.

Gradients coming from the policy LSTM are blocked and only gradients originating from the Q-network LSTM are allowed to back-propagate into the convolutional neural network. We block gradients from the policy head for increased stability, as this avoids positive feedback loops between $\pi$ and $q_i$ caused by shared representations. We used the Adam optimiser (Kingma & Ba, 2014),

---

[4]All results are reported with respect to the combined total number of observations obtained over all worker machines.

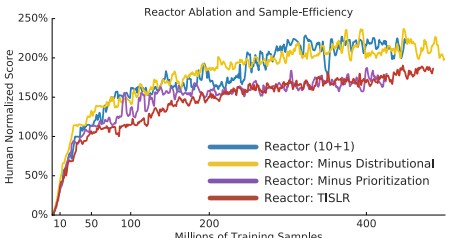 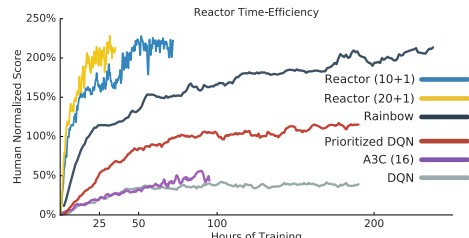

Figure 3: (Left) Reactor performance as various components are removed. (Right) Performance comparison as a function of training time in hours. Rainbow learning curve provided by Hessel et al. (2017).

with a learning rate of $5 \times 10^{-5}$ and zero momentum because asynchronous updates induce implicit momentum (Mitliagkas et al., 2016). Further discussion of hyperparameters and their optimization can be found in Appendix 6.1.

## 4 EXPERIMENTAL RESULTS

We trained and evaluated Reactor on 57 Atari games (Bellemare et al., 2013). Figure 3 compares the performance of Reactor with different versions of Reactor each time leaving one of the algorithmic improvements out. We can see that each of the algorithmic improvements (Distributional retrace, beta-LOO and prioritized replay) contributed to the final results. While prioritization was arguably the most important component, Beta-LOO clearly outperformed TISLR algorithm. Although distributional and non-distributional versions performed similarly in terms of median human normalized scores, distributional version of the algorithm generalized better when tested with random human starts (Table 1).

| ALGORITHM | NORMALIZED SCORES | MEAN RANK | ELO |
|---|---|---|---|
| RANDOM | 0.00 | 11.65 | -563 |
| HUMAN | 1.00 | 6.82 | 0 |
| DQN | 0.69 | 9.05 | -172 |
| DDQN | 1.11 | 7.63 | -58 |
| DUEL | 1.17 | 6.35 | 32 |
| PRIOR | 1.13 | 6.63 | 13 |
| PRIOR. DUEL. | 1.15 | 6.25 | 40 |
| A3C LSTM | 1.13 | 6.30 | 37 |
| RAINBOW | 1.53 | 4.18 | 186 |
| REACTOR ND [5] | 1.51 | 4.98 | 126 |
| REACTOR | 1.65 | 4.58 | 156 |
| REACTOR 500M | **1.82** | **3.65** | **227** |

Table 1: Random human starts

| ALGORITHM | NORMALIZED SCORES | MEAN RANK | ELO |
|---|---|---|---|
| RANDOM | 0.00 | 10.93 | -673 |
| HUMAN | 1.00 | 6.89 | 0 |
| DQN | 0.79 | 8.65 | -167 |
| DDQN | 1.18 | 7.28 | -27 |
| DUEL | 1.51 | 5.19 | 143 |
| PRIOR | 1.24 | 6.11 | 70 |
| PRIOR. DUEL. | 1.72 | 5.44 | 126 |
| ACER[6] 500M | 1.9 | - | - |
| RAINBOW | **2.31** | 3.63 | 270 |
| REACTOR ND [5] | 1.80 | 4.53 | 195 |
| REACTOR | 1.87 | 4.46 | 196 |
| REACTOR 500M | 2.30 | **3.47** | **280** |

Table 2: 30 random no-op starts.

### 4.1 COMPARING TO PRIOR WORK

We evaluated Reactor with target update frequency $T_{update} = 1000$, $\lambda = 1.0$ and $\beta$-LOO with $\beta = 1$ on 57 Atari games trained on 10 machines in parallel. We averaged scores over 200 episodes using 30 random human starts and noop starts (Tables 4 and 5 in the Appendix). We calculated mean and median human normalised scores across all games. We also ranked all algorithms (including random and human scores) for each game and evaluated mean rank of each algorithm across all 57 Atari games. We also evaluated mean Rank and Elo scores for each algorithm for both human and noop start settings. Please refer to Section 6.2 in the Appendix for more details.

Tables 1 & 2 compare versions of our algorithm,[5] with several other state-of-art algorithms across 57 Atari games for a fixed random seed across all games (Bellemare et al., 2013). We compare Reactor

---

[5] 'ND' stands for a non-distributional (i.e. classical) version of Reactor using Retrace (Munos et al., 2016).

against are: DQN (Mnih et al., 2015), Double DQN (Van Hasselt et al., 2016), DQN with prioritised experience replay (Schaul et al., 2015), dueling architecture and prioritised dueling (Wang et al., 2015), ACER (Wang et al., 2017), A3C (Mnih et al., 2016), and Rainbow (Hessel et al., 2017). Each algorithm was exposed to 200 million frames of experience, or 500 million frames when followed by 500M, and the same pre-processing pipeline including 4 action repeats was used as in the original DQN paper (Mnih et al., 2015).

In Table 1, we see that Reactor exceeds the performance of all algorithms across all metrics, despite requiring under two days of training. With 500 million frames and four days training we see Reactor's performance continue to improve significantly. The difference in time-efficiency is especially apparent when comparing Reactor and Rainbow (see Figure 3, right). Additionally, unlike Rainbow, Reactor does not use Noisy Networks (Fortunato et al., 2017), which was reported to have contributed to the performance gains. When evaluating under the no-op starts regime (Table 2), Reactor out performs all methods except for Rainbow. This suggests that Rainbow is more sample-efficient when training and evaluation regimes match exactly, but may be overfitting to particular trajectories due to the significant drop in performance when evaluated on the random human starts.

Regarding ACER, another Retrace-based actor-critic architecture, both classical and distributional versions of Reactor (Figure 3) exceeded the best reported median human normalized score of 1.9 with noop starts achieved in 500 million steps.[6]

## 5 CONCLUSION

In this work we presented a new off-policy agent based on Retrace actor-critic architecture and show that it achieves similar performance as the current state-of-the-art while giving significant real-time performance gains. We demonstrate the benefits of each of the suggested algorithmic improvements, including Distributional Retrace, beta-LOO policy gradient and contextual priority tree.

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

## 6 APPENDIX

**Proposition 1.** *Assume $\hat{a} \sim \mu$ and that $\mathbb{E}[R(\hat{a})] = Q^\pi(\hat{a})$. Then, the bias of $G_{\beta\text{-LOO}}$ is $\left| \sum_a (1 - \mu(a)\beta(a))\nabla\pi(a)[Q(a) - Q^\pi(a)] \right|$.*

*Proof.* The bias of $\hat{G}_{\beta\text{-LOO}}$ is

$$
\begin{aligned}
\mathbb{E}[\hat{G}_{\beta\text{-LOO}}] - G &= \sum_a \mu(a)[\beta(a)(\mathbb{E}[R(a)] - Q(a))]\nabla\pi(a) + \sum_a Q(a)\nabla\pi(a) - G \\
&= \sum_a (1 - \mu(a)\beta(a))[Q(a) - Q^\pi(a)]\nabla\pi(a)
\end{aligned}
$$

$\square$

### 6.1 HYPERPARAMETER OPTIMIZATION

As we believe that algorithms should be robust with respect to the choice of hyperparameters, we spent little effort on parameter optimization. In total, we explored three distinct values of learning rates and two values of ADAM momentum (the default and zero) and two values of $T_{update}$ on a subset of 7 Atari games without prioritization using non-distributional version of Reactor. We later used those values for all experiments. We did not optimize for batch sizes and sequence length or any prioritization hyperparamters.

### 6.2 RANK AND ELO EVALUATION

Commonly used mean and median human normalized scores have several disadvantages. A mean human normalized score implicitly puts more weight on games that computers are good and humans are bad at. Comparing algorithm by a mean human normalized score across 57 Atari games is almost equivalent to comparing algorithms on a small subset of games close to the median and thus dominating the signal. Typically a set of ten most score-generous games, namely Assault, Asterix, Breakout, Demon Attack, Double Dunk, Gopher, Pheonix, Stargunner, Up'n Down and Video Pinball can explain more than half of inter-algorithm variance. A median human normalized score has the opposite disadvantage by effectively discarding very easy and very hard games from the comparison. As typical median human normalized scores are within the range of 1-2.5, an algorithm which scores zero points on Montezuma's Revenge is evaluated equal to the one which scores 2500 points, as both performance levels are still below human performance making incremental improvements on hard games not being reflected in the overall evaluation. In order to address both problem, we also evaluated mean rank and Elo metrics for inter-algorithm comparison. Those metrics implicitly assign the same weight to each game, and as a result is more sensitive of relative performance on very hard and easy games: swapping scores of two algorithms on any game would result in the change of both mean rank and Elo metrics.

We calculated separate mean rank and Elo scores for each algorithm using results of test evaluations with 30 random noop-starts and 30 random human starts (Tables 5 and 4). All algorithms were ranked across each game separately, and a mean rank was evaluated across 57 Atari games. For Elo score evaluation algorithm, $A$ was considered to win over algorithm $B$ if it obtained more scores on a given Atari. We produced an empirical win-probability matrix by summing wins across all games and used this matrix to evaluate Elo scores. A ranking difference of 400 corresponds to the odds of winning of 10:1 under the Gaussian assumption.

### 6.3 CONTEXTUAL PRIORITY TREE

Contextual priority tree is one possible implementation of lazy prioritization (Figure 4). All sequence keys are put into a balanced binary search tree which maintains a temporal order. An AVL tree (Velskii & Landis (1976)) was chosen due to the ease of implementation and because it is on average more evenly balanced than a Red-Black Tree.

Each tree node has up to two children (left and right) and contains currently stored key and a priority of the key which is either set or is unknown. Some trees may only have a single child subtree while

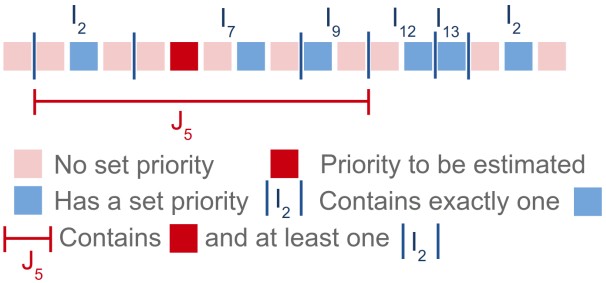

Figure 4: Illustration of Lazy prioritization, where sequences with no explicitly assigned priorities get priorities estimated by a linear combination of nearby assigned priorities. Exact boundaries of blue and red intervals are arbitrary (as long as all conditions described in Section 3.3 are satisfied) thus leading to many possible algorithms. Each square represents an individual sequence of size 32 (sequences overlap). Inverse sizes of blue regions work as local density estimates allowing to produce unbiased priority estimates.

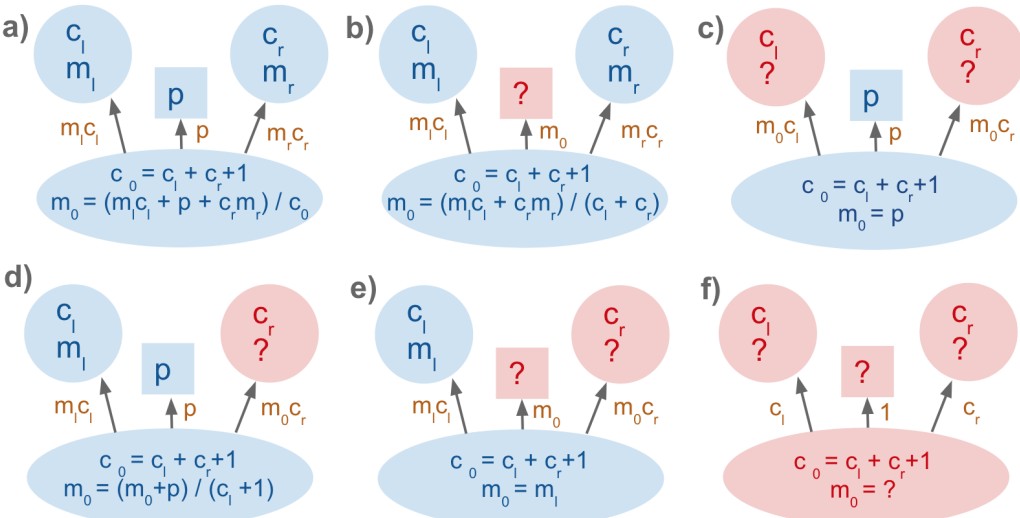

Figure 5: Rules used to evaluate summary statistics on each node of a binary search tree where all sequence keys are kept sorted by temporal order. $c_l$ and $c_r$ are total number of nodes within left and right subtrees. $m_l$ and $m_l$ are estimated mean priorities per node within the subtree. A central square node corresponds to a single key stored within the parent node with its corresponding priority of $p$ (if set) or ? if not set. Red subtrees do not have any singe child with a set priority, and a result do not have priority estimates. A red square shows that priority of the key stored within the parent node is not known. Unknown mean priorities is marked by a question mark. Empty child nodes simply behave as if $c = 0$ with $p =$?. Rules a-f illustrate how mean values are propagated down from children to parents when priorities are only partially known (rules d and e also apply symmetrically). Sampling is done by going from the root node up the tree by selecting one of the children (or the current key) stochastically proportional to orange proportions. Sampling terminates once the current (square) key is chosen.

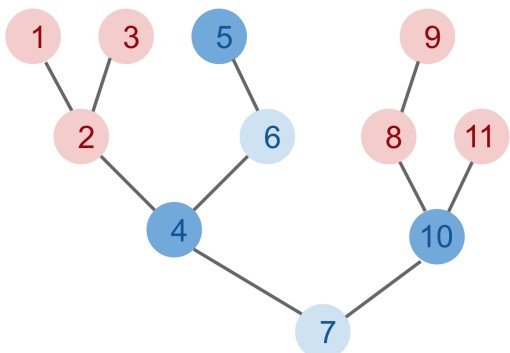

Figure 6: Example of a balanced priority tree. Dark blue nodes contain keys with known priorities, light blue nodes have at least one child with at least a single known priority, while ping nodes do not have any priority estimates. Nodes 1, 2 and 3 will obtain priority estimates equal to $2/3$ of the priority of key 5 and $1/3$ of the priority of node 4. This implies that estimated priorities of keys 1, 2 and 3 are implicitly defined by keys 4 and 6. Nodes 8, 9 and 11 are estimated to have the same priority as node 10.

some may have none. In addition to this information, we were tracking other summary statistics at each node which was re-evaluated after each tree rotation. The summary statistics was evaluated by consuming previously evaluated summary statistics of both children and a priority of the key stored within the current node. In particular, we were tracking a total number of nodes within each subtree and mean-priority estimates updated according to rules shown in Figure 5. The total number of nodes within each subtree was always known ($c$ in Figure 5), while mean priority estimates per key ($m$ in Figure 5) could either be known or unknown.

If a mean priority of either one child subtree or a key stored within the current node is unknown then it can be estimated to by exploiting information coming from another sibling subtree or a priority stored within the parent node.

Sampling was done by traversing the tree from the root node up while sampling either one of the children subtrees or the currently held key proportionally to the total estimated priority masses contained within. The rules used to evaluate proportions are shown in orange in Figure 5. Similarly, probabilities of arbitrary keys can be queried by traversing the tree from the root node towards the child node of an interest while maintaining a product of probabilities at each branching point. Insertion, deletion, sampling and probability query operations can be done in O(ln(n)) time.

The suggested algorithm has the desired property that it becomes a simple proportional sampling algorithm once all the priorities are known. While some key priorities are unknown, they are estimated by using nearby known key priorities (Figure 6).

Each time when a new sequence key is added to the tree, it was set to have an unknown priority. Any priority was assigned only after the key got first sampled and the corresponding sequence got passed through the learner. When a priority of a key is set or updated, the key node is deliberately removed from and placed back to the tree in order to become a leaf-node. This helped to set priorities of nodes in the immediate vicinity more accurately by using the freshest information available.

## 6.4 NETWORK ARCHITECTURE

The value of $\epsilon = 0.01$ is the minimum probability of choosing a random action and it is hard-coded into the policy network. Figure 7 shows the overall network topology while Table 3 specifies network layer sizes.

Action value estimate

Q(x, a)

V(x)  A(x, a)  Current policy  π(x, a)

LSTM  LSTM

Linear

Convnet

Figure 7: Network architecture.

Table 3: Specification of the neural network used (illustrated in Figure 7)

| LAYER | INPUT SIZE | PARAMETERS | | |
|---|---|---|---|---|
| CONVOLUTIONAL | | KERNEL WIDTH | OUTPUT CHANNELS | STRIDES |
| CONV 1 | [84, 84, 1] | [8, 8] | 16 | 4 |
| CONCATRELU | [20, 20, 16] | | | |
| CONV 2 | [20, 20, 32] | [4, 4] | 32 | 2 |
| CONCATRELU | [9, 9, 32] | | | |
| CONV 3 | [9, 9, 64] | [3, 3] | 32 | 1 |
| CONCATRELU | [7, 7, 32] | | | |
| FULLY CONNECTED | | OUTPUT SIZE | | |
| LINEAR | [7, 7, 64] | 128 | | |
| CONCATRELU | [128] | | | |
| RECURRENT | | OUTPUT SIZE | | |
| $\pi$ | | | | |
| LSTM | [256] | 128 | | |
| LINEAR | [128] | 32 | | |
| CONCATRELU | [32] | | | |
| LINEAR | [64] | #ACTIONS | | |
| SOFTMAX | [#ACTIONS] | #ACTIONS | | |
| X$(1-\epsilon)+\epsilon$/#ACTIONS | [#ACTIONS] | #ACTIONS | | |
| RECURRENT $Q$ | | OUTPUT SIZE | | |
| LSTM | [256] | 128 | | |
| VALUE LOGIT HEAD | | OUTPUT SIZE | | |
| LINEAR | [128] | 32 | | |
| CONCATRELU | [32] | | | |
| LINEAR | [64] | #BINS | | |
| ADVANTAGE LOGIT HEAD | | #ACTIONS $\times$ #BINS | | |
| LINEAR | [128] | 32 | | |
| CONCATRELU | [32] | | | |

## 6.5 Comparisons with Rainbow

In this section we compare Reactor with the recently published Rainbow agent (Hessel et al., 2017). While ACER is the most closely related algorithmically, Rainbow is most closely related in terms of performance and thus a deeper understanding of the trade-offs between Rainbow and Reactor may benefit interested readers. There are many architectural and algorithmic differences between Rainbow and Reactor. We will therefore begin by highlighting where they agree. Both use a categorical action-value distribution critic (Bellemare et al., 2017), factored into state and state-action logits (Wang et al., 2015),

$$q_i(x,a) = \frac{l_i(x,a)}{\sum_j l_j(x,a)}, \quad l_i(x,a) = l_i(x) + l_i(x,a) - \frac{1}{|\mathcal{A}|} \sum_{b \in \mathcal{A}} l_i(x,b).$$

Both use prioritized replay, and finally, both perform $n$-step Bellman updates.

Despite these similarities, Reactor and Rainbow are fundamentally different algorithms and are based upon different lines of research. While Rainbow uses Q-Learning and is based upon DQN (Mnih et al., 2015), Reactor is an actor-critic algorithm most closely based upon A3C (Mnih et al., 2016). Each inherits some design choices from their predecessors, and we have not performed an extensive ablation comparing these various differences. Instead, we will discuss four of the differences we believe are important but less obvious.

First, the network structures are substantially different. Rainbow uses noisy linear layers and ReLU activations throughout the network, whereas Reactor uses standard linear layers and concatenated ReLU activations throughout. To overcome partial observability, Rainbow, inheriting this choice from DQN, uses *frame stacking*. On the other hand, Reactor, inheriting its choice from A3C, uses LSTMs after the convolutional layers of the network. It is also difficult to directly compare the number of parameters in each network because the use of noisy linear layers doubles the number of parameters, although half of these are used to control noise, while the LSTM units in Reactor require more parameters than a corresponding linear layer would.

Second, both algorithms perform $n$-step updates, however, the Rainbow $n$-step update does not use any form of off-policy correction. Because of this, Rainbow is restricted to using only small values of $n$ (e.g. $n = 3$) because larger values would make sequences more off-policy and hurt performance. By comparison, Reactor uses our proposed distributional Retrace algorithm for off-policy correction of $n$-step updates. This allows the use of larger values of $n$ (e.g. $n = 33$) without loss of performance.

Third, while both agents use prioritized replay buffers (Schaul et al., 2016), they each store different information and prioritize using different algorithms. Rainbow stores a tuple containing the state $x_{t-1}$, action $a_{t-1}$, sum of $n$ discounted rewards $\sum_{k=0}^{n-1} r_{t+k} \prod_{m=0}^{k-1} \gamma_{t+m}$, product of $n$ discount factors $\prod_{k=0}^{n-1} \gamma_{t+k}$, and next-state $n$ steps away $x_{t+n-1}$. Tuples are prioritized based upon the last observed TD error, and inserted into replay with a maximum priority. Reactor stores length $n$ sequences of tuples $(x_{t-1}, a_{t-1}, r_t, \gamma_t)$ and also prioritizes based upon the observed TD error. However, when inserted into the buffer the priority is instead inferred based upon the known priorities of neighboring sequences. This priority inference was made efficient using the previously introduced contextual priority tree, and anecdotally we have seen it improve performance over a simple maximum priority approach.

Finally, the two algorithms have different approaches to exploration. Rainbow, unlike DQN, does not use $\epsilon$-greedy exploration, but instead replaces all linear layers with noisy linear layers which induce randomness throughout the network. This method, called Noisy Networks (Fortunato et al., 2017), creates an adaptive exploration integrated into the agent's network. Reactor does not use noisy networks, but instead uses the same entropy cost method used by A3C and many others (Mnih et al., 2016), which penalizes deterministic policies thus encouraging indifference between similarly valued actions. Because Rainbow can essentially learn not to explore, it may learn to become entirely greedy in the early parts of the episode, while still exploring in states not as frequently seen. In some sense, this is precisely what we want from an exploration technique, but it may also lead to highly deterministic trajectories in the early part of the episode and an increase in overfitting to those trajectories. We hypothesize that this may be the explanation for the significant difference in Rainbow's performance between evaluation under no-op and random human starts, and why Reactor does not show such a large difference.

## 6.6 ATARI RESULTS

Table 4: Scores for each game evaluated with 30 random human starts. Reactor was evaluated by averaging scores over 200 episodes. All scores (except for Reactor) were taken from Wang et al. (2015), Mnih et al. (2016) and Hessel et al. (2017).

| GAME AGENT | RANDOM | HUMAN | DQN | DDQN | DUEL | PRIOR | PRIOR. DUEL. | A3C LSTM | RAINBOW | REACTOR ND [5] | REACTOR | REACTOR 500M |
|---|---|---|---|---|---|---|---|---|---|---|---|---|
| ALIEN | 128.3 | **6371.3** | 634.0 | 1033.4 | 1486.5 | 1334.7 | 823.7 | 945.3 | 6022.9 | 924.1 | 2763.9 | 4958.6 |
| AMIDAR | 11.8 | **1540.4** | 178.4 | 169.1 | 172.7 | 129.1 | 238.4 | 173.0 | 202.8 | 174.8 | 423.0 | 503.2 |
| ASSAULT | 166.9 | 628.9 | 3489.3 | 6060.8 | 3994.8 | 6548.9 | 10950.6 | 14497.9 | 14491.7 | **15041.2** | 9994.2 | 6713.8 |
| ASTERIX | 164.5 | 7536.0 | 3170.5 | 16837.0 | 15840.0 | 22484.5 | **364200.0** | 17244.5 | 280114.0 | 7269.8 | 26240.8 | 191948.0 |
| ASTEROIDS | 871.3 | **36517.3** | 1458.7 | 1193.2 | 2035.4 | 1745.1 | 1021.9 | 5093.1 | 2249.4 | 3764.4 | 2771.3 | 4215.0 |
| ATLANTIS | 13463.0 | 26575.0 | 292491.0 | 319688.0 | 445360.0 | 330647.0 | 423252.0 | 875822.0 | 814684.0 | **897803.5** | 280809.5 | 274196.0 |
| BANK HEIST | 21.7 | 644.5 | 312.7 | 886.0 | **1129.3** | 876.6 | 1004.6 | 932.8 | 826.0 | 852.8 | 710.1 | 876.0 |
| BATTLEZONE | 3560.0 | 33030.0 | 23750.0 | 24740.0 | 31320.0 | 25520.0 | 30650.0 | 20760.0 | 52040.0 | **84985.0** | 52300.0 | 57075.0 |
| BEAM RIDER | 254.6 | 14961.0 | 9743.2 | 17417.2 | 14591.3 | 31181.3 | **37412.2** | 24622.2 | 21768.5 | 9298.9 | 9941.6 | 12877.0 |
| BERZERK | 196.1 | **2237.5** | 493.4 | 1011.1 | 910.6 | 865.9 | 2178.6 | 862.2 | 1793.4 | 1221.3 | 1186.0 | 1643.4 |
| BOWLING | 35.2 | **146.5** | 56.5 | 69.6 | 65.7 | 52.0 | 50.4 | 41.8 | 39.4 | 63.7 | 72.4 | 76.0 |
| BOXING | -1.5 | 9.6 | 70.3 | 73.5 | 77.3 | 72.3 | **79.2** | 37.3 | 54.9 | 60.2 | 75.1 | 55.6 |
| BREAKOUT | 1.6 | 27.9 | 354.5 | 368.9 | 411.6 | 343.0 | 354.6 | **766.8** | 379.5 | 459.0 | 465.0 | 478.4 |
| CENTIPEDE | 1925.5 | **10321.9** | 3973.9 | 3853.5 | 4881.0 | 3489.1 | 5570.2 | 1997.0 | 7160.9 | 3974.5 | 2584.0 | 2674.3 |
| CHOPPER COMMAND | 644.0 | 8930.0 | 5017.0 | 3495.0 | 3784.0 | 4635.0 | 8058.0 | 10150.0 | 10916.0 | 17312.0 | 33150.0 | **71442.5** |
| CRAZY CLIMBER | 9337.0 | 32667.0 | 98128.0 | 113782.0 | 124566.0 | 127512.0 | 127853.0 | 138518.0 | 143962.0 | 151295.0 | **182399.0** | 209784.0 |
| DEFENDER | 1965.5 | 14296.0 | 15917.5 | 27510.0 | 33996.0 | 23666.5 | 34415.0 | **233021.5** | 47671.3 | 162327.5 | 110446.3 | 221671.0 |
| DEMON ATTACK | 208.3 | 3442.8 | 12550.7 | 69803.4 | 56322.8 | 61277.5 | 73371.3 | 115201.9 | 109670.7 | **120682.2** | 101435.4 | 113853.0 |
| DOUBLE DUNK | -16.0 | -14.4 | -6.0 | -0.3 | -0.8 | 16.0 | -10.7 | 0.1 | -0.6 | **22.2** | 11.0 | 22.0 |
| ENDURO | -81.8 | 740.2 | 626.7 | 1216.6 | 2077.4 | 1831.0 | **2223.9** | -82.5 | 2061.1 | 2054.3 | 2127.0 | 2138.3 |
| FISHING DERBY | -77.1 | 5.1 | -1.6 | 3.2 | -4.1 | 9.8 | 17.0 | 22.6 | 22.6 | **23.2** | 17.7 | 21.7 |
| FREEWAY | 0.1 | 25.6 | 26.9 | 28.8 | 0.2 | 28.9 | 28.2 | 0.1 | **29.1** | 19.4 | 26.9 | 26.9 |
| FROSTBITE | 66.4 | 4202.8 | 496.1 | 1448.1 | 2332.4 | 3510.0 | 4038.4 | 197.6 | 4141.1 | 3497.8 | 4358.7 | **4743.5** |
| GOPHER | 250.0 | 2311.0 | 8190.4 | 15253.0 | 20051.4 | 34858.8 | **105148.4** | 17106.8 | 72595.7 | 36286.2 | 60743.2 | 89306.8 |
| GRAVITAR | 245.5 | **3116.0** | 298.0 | 200.5 | 297.0 | 269.5 | 167.0 | 320.0 | 567.5 | 544.0 | 583.0 | 779.8 |
| H.E.R.O. | 1580.3 | 25839.4 | 14992.9 | 14892.5 | 15207.9 | 20889.9 | 15459.2 | 28889.5 | **50496.8** | 22673.3 | 30253.7 | 37833.4 |
| ICE HOCKEY | -9.7 | 0.5 | -1.6 | -2.5 | -1.3 | -0.2 | 0.5 | -1.7 | -0.7 | **11.1** | 0.9 | 4.9 |
| JAMES BOND 007 | 33.5 | 368.5 | 697.5 | 573.0 | 835.5 | 3961.0 | 585.0 | 613.0 | **18142.3** | 12655.3 | 6741.0 | 13987.5 |
| KANGAROO | 100.0 | 2739.0 | 4496.0 | 11204.0 | 10334.0 | **12185.0** | 861.0 | 125.0 | 10841.0 | 9111.0 | 5143.5 | 5587.5 |
| KRULL | 1151.9 | 2109.1 | 6206.0 | 6796.1 | **8051.6** | 6872.8 | 7658.6 | 5911.4 | 6715.5 | 7450.1 | 7815.9 | 7621.8 |
| KUNG-FU MASTER | 304.0 | 20786.8 | 20882.0 | 30207.0 | 24288.0 | 31676.0 | 37484.0 | 40835.0 | 28999.8 | 48781.5 | 49767.9 | **55357.7** |
| MONTEZUMA'S REVENGE | 25.0 | **4182.0** | 47.0 | 42.0 | 22.0 | 51.0 | 24.0 | 41.0 | 154.0 | 45.0 | 984.0 | 1045.5 |
| MS. PAC-MAN | 197.8 | **15375.0** | 1092.3 | 1241.3 | 2250.6 | 1865.9 | 1007.8 | 850.7 | 2570.2 | 1102.4 | 1714.6 | 2540.1 |
| NAME THIS GAME | 1747.8 | 6796.0 | 6738.8 | 8960.3 | 11185.1 | 10497.6 | **13637.9** | 12093.7 | 11686.5 | 11359.7 | 8288.8 | 8207.7 |
| PHOENIX | 1134.4 | 6686.2 | 7484.8 | 12366.5 | 20410.5 | 16903.6 | 63597.0 | 74786.7 | **103061.6** | 8650.4 | 34956.5 | 41426.7 |
| PITFALL! | -348.8 | **5998.9** | -113.2 | -186.7 | -46.9 | -427.0 | -243.6 | -135.7 | -37.6 | -99.2 | -138.8 | -146.9 |
| PONG | -18.0 | 15.5 | 18.0 | **19.1** | 18.8 | 18.9 | 18.4 | 10.7 | 19.0 | 18.5 | 17.9 | 18.0 |
| PRIVATE EYE | 662.8 | **64169.1** | 207.9 | -575.5 | 292.6 | 670.7 | 1277.6 | 421.1 | 1704.4 | 5638.1 | 5751.4 | 5309.4 |
| Q*BERT | 183.0 | 12085.0 | 9271.5 | 11020.8 | 14175.8 | 9944.0 | 14063.0 | **21307.5** | 18397.6 | 18117.8 | 20689.3 | 19505.8 |
| RIVER RAID | 588.3 | 14382.2 | 4748.5 | 10838.4 | **16569.4** | 11807.2 | 16496.8 | 6591.9 | 15608.1 | 12074.0 | 11987.3 | 13032.9 |
| ROAD RUNNER | 200.0 | 6878.0 | 35215.0 | 43156.0 | 58549.0 | 52264.0 | 54630.0 | 73949.0 | 54261.0 | 58568.5 | 66247.0 | **97663.5** |
| ROBOTANK | 2.4 | 8.9 | 58.5 | 59.1 | 62.0 | 56.2 | 24.7 | 2.6 | 55.2 | **67.1** | 63.7 | 64.2 |
| SEAQUEST | 215.5 | **40425.8** | 4216.7 | 14498.0 | 37361.6 | 25463.7 | 1431.2 | 1326.1 | 19176.0 | 3802.3 | 10694.0 | 25887.9 |
| SKIING | -15287.4 | **-3686.6** | -12142.1 | -11490.4 | -11928.0 | -10169.1 | -18955.8 | -14863.8 | -11685.8 | -9461.8 | -11809.5 | -11904.4 |
| SOLARIS | 2047.2 | **11032.6** | 1295.4 | 810.0 | 1768.4 | 2272.8 | 280.6 | 1936.4 | 2860.7 | 2000.1 | 1740.7 | 1728.3 |
| SPACE INVADERS | 182.6 | 1464.9 | 1293.8 | 2628.7 | 5993.1 | 3912.1 | 8978.0 | **23846.0** | 12629.0 | 1240.9 | 2010.2 | 7950.9 |
| STARGUNNER | 697.0 | 9528.0 | 52970.0 | 58365.0 | 90804.0 | 61582.0 | 127073.0 | **164766.0** | 123853.0 | 47532.0 | 67384.5 | 74451.5 |
| SURROUND | -9.7 | 5.4 | -6.0 | 1.9 | 4.0 | 5.9 | -0.2 | -8.3 | **7.0** | 0.3 | 5.1 | 5.5 |
| TENNIS | -21.4 | -6.7 | 11.1 | -7.8 | 4.4 | -5.3 | -13.2 | -6.4 | -2.2 | 22.5 | 22.1 | **22.8** |
| TIME PILOT | 3273.0 | 5650.0 | 4786.0 | 6608.0 | 6601.0 | 5963.0 | 4871.0 | **27202.0** | 11190.5 | 15860.5 | 17205.0 | 16929.5 |
| TUTANKHAM | 12.7 | 138.3 | 45.6 | 92.2 | 48.0 | 56.9 | 108.6 | 144.2 | 126.9 | 162.3 | **164.6** | 162.9 |
| UP'N DOWN | 707.2 | 9896.1 | 8038.5 | 19086.9 | 24759.2 | 12157.4 | 22681.3 | 105728.7 | 92640.6 | **123798.7** | 48148.6 | 49310.4 |
| VENTURE | 18.0 | **1039.0** | 136.0 | 21.0 | 200.0 | 94.0 | 29.0 | 25.0 | 45.0 | 26.0 | 782.0 | 810.0 |
| VIDEO PINBALL | 20452.0 | 15641.1 | 154414.1 | 367823.7 | 110976.2 | 295972.8 | 447408.6 | 470310.5 | 506817.2 | 273249.0 | 484384.2 | **538975.0** |
| WIZARD OF WOR | 804.0 | 4556.0 | 1609.0 | 6201.0 | 7054.0 | 5727.0 | 10471.0 | **18082.0** | 14631.5 | 15265.5 | 10995.0 | 16731.5 |
| YARS' REVENGE | 1476.9 | 47135.2 | 4577.5 | 6270.6 | 25976.5 | 4687.4 | 58145.9 | 5615.5 | 93007.9 | 87928.9 | 82113.5 | **129871.0** |
| ZAXXON | 475.0 | 8443.0 | 4412.0 | 8593.0 | 10164.0 | 9474.0 | 11320.0 | 23519.0 | 19658.0 | 14155.5 | 20644.0 | **25147.5** |

Table 5: Scores for each game evaluated with 30 random noop starts. Reactor was evaluated by averaging scores over 200 episodes. All scores (except for Reactor) were taken from Wang et al. (2015) and Hessel et al. (2017).

| GAME AGENT | RANDOM | HUMAN | DQN | DDQN | DUEL | PRIOR | PRIOR. DUEL. | RAINBOW | REACTOR ND [5] | REACTOR | REACTOR 500M |
|---|---|---|---|---|---|---|---|---|---|---|---|
| ALIEN | 227.8 | 7127.7 | 1620.0 | 3747.7 | 4461.4 | 4203.8 | 3941.0 | 9491.7 | 4199.4 | 6482.1 | **12689.1** |
| AMIDAR | 5.8 | 1719.5 | 978.0 | 1793.3 | 2354.5 | 1838.9 | 2296.8 | **5131.2** | 1546.8 | 833.0 | 1015.8 |
| ASSAULT | 222.4 | 742.0 | 4280.4 | 5393.2 | 4621.0 | 7672.1 | 11477.0 | 14198.5 | **17543.8** | 11013.5 | 8323.3 |
| ASTERIX | 210.0 | 8503.3 | 4359.0 | 17356.5 | 28188.0 | 31527.0 | 375080.0 | **428200.3** | 16121.0 | 36238.5 | 205914.0 |
| ASTEROIDS | 719.1 | **47388.7** | 1364.5 | 734.7 | 2837.7 | 2654.3 | 1192.7 | 2712.8 | 4467.4 | 2780.4 | 3726.1 |
| ATLANTIS | 12850.0 | 29028.1 | 279987.0 | 106056.0 | 382572.0 | 357324.0 | 395762.0 | 826659.5 | **968179.5** | 308258.0 | 302831.0 |
| BANK HEIST | 14.2 | 753.1 | 455.0 | 1030.6 | **1611.9** | 1054.6 | 1503.1 | 1358.0 | 1236.8 | 988.7 | 1259.7 |
| BATTLEZONE | 2360.0 | 37187.5 | 29900.0 | 31700.0 | 37150.0 | 31530.0 | 35520.0 | 62010.0 | **98235.0** | 61220.0 | 64070.0 |
| BEAM RIDER | 363.9 | 16926.5 | 8627.5 | 13772.8 | 12164.0 | 23384.2 | **30276.5** | 16850.2 | 8811.8 | 8566.5 | 11033.4 |
| BERZERK | 123.7 | 2630.4 | 585.6 | 1225.4 | 1472.6 | 1305.6 | **3409.0** | 2545.6 | 1515.7 | 1641.4 | 2303.1 |
| BOWLING | 23.1 | **160.7** | 50.4 | 68.1 | 65.5 | 47.9 | 46.7 | 30.0 | 59.3 | 75.4 | 81.0 |
| BOXING | 0.1 | 12.1 | 88.0 | 91.6 | 99.4 | 95.6 | 98.9 | 99.6 | **99.7** | 99.4 | 99.4 |
| BREAKOUT | 1.7 | 30.5 | 385.5 | 418.5 | 345.3 | 373.9 | 366.0 | 417.5 | 509.5 | **518.4** | 514.8 |
| CENTIPEDE | 2090.9 | **12017.0** | 4657.7 | 5409.4 | 7561.4 | 4463.2 | 7687.5 | 8167.3 | 7267.2 | 3402.8 | 3422.0 |
| CHOPPER COMMAND | 811.0 | 7387.8 | 6126.0 | 5809.0 | 11215.0 | 8600.0 | 13185.0 | 16654.0 | 19901.5 | 37568.0 | **107779.0** |
| CRAZY CLIMBER | 10780.5 | 35829.4 | 110763.0 | 117282.0 | 143570.0 | 141161.0 | 162224.0 | 168788.5 | 173274.0 | 194347.0 | **236422.0** |
| DEFENDER | 2874.5 | 18688.9 | 23633.0 | 35338.5 | 42214.0 | 31286.5 | 41324.5 | 55105.0 | 181074.3 | 113128.0 | **223025.0** |
| DEMON ATTACK | 152.1 | 1971.0 | 12149.4 | 58044.2 | 60813.3 | 71846.4 | 72878.6 | 111185.2 | **122782.5** | 100189.0 | 115154.0 |
| DOUBLE DUNK | -18.6 | -16.4 | -6.6 | -5.5 | 0.1 | 18.5 | -12.5 | -0.3 | 23.0 | 11.4 | **23.0** |
| ENDURO | 0.0 | 860.5 | 729.0 | 1211.8 | 2258.2 | 2093.0 | **2306.4** | 2125.9 | 2211.3 | 2230.1 | 2224.2 |
| FISHING DERBY | -91.7 | -38.7 | -4.9 | 15.5 | **46.4** | 39.5 | 41.3 | 31.3 | 33.1 | 23.2 | 30.4 |
| FREEWAY | 0.0 | 29.6 | 30.8 | 33.3 | 0.0 | 33.7 | 33.0 | **34.0** | 22.3 | 31.4 | 31.5 |
| FROSTBITE | 65.2 | 4334.7 | 797.4 | 1683.3 | 4672.8 | 4380.1 | 7413.0 | **9590.5** | 7136.7 | 8042.1 | 7932.2 |
| GOPHER | 257.6 | 2412.5 | 8777.4 | 14840.8 | 15718.4 | 32487.2 | **104368.2** | 70354.6 | 36279.1 | 69135.1 | 89851.0 |
| GRAVITAR | 173.0 | **3351.4** | 473.0 | 412.0 | 588.0 | 548.5 | 238.0 | 1419.3 | 1804.8 | 1073.8 | 2041.8 |
| H.E.R.O. | 1027.0 | 30826.4 | 20437.8 | 20130.2 | 20818.2 | 23037.7 | 21036.5 | **55887.4** | 27833.0 | 35542.2 | 43360.4 |
| ICE HOCKEY | -11.2 | 0.9 | -1.9 | -2.7 | 0.5 | 1.3 | -0.4 | 1.1 | **15.7** | 3.4 | 10.7 |
| JAMES BOND 007 | 29.0 | 302.8 | 768.5 | 1358.0 | 1312.5 | 5148.0 | 812.0 | 19809.0 | 14524.0 | 7869.2 | 16056.2 |
| KANGAROO | 52.0 | 3035.0 | 7259.0 | 12992.0 | 14854.0 | **16200.0** | 1792.0 | 14637.5 | 13349.0 | 10484.5 | 11266.5 |
| KRULL | 1598.0 | 2665.5 | 8422.3 | 7920.5 | **11451.9** | 9728.0 | 10374.4 | 8741.5 | 10237.8 | 9930.8 | 9896.0 |
| KUNG-FU MASTER | 258.5 | 22736.3 | 26059.0 | 29710.0 | 34294.0 | 39581.0 | 48375.0 | 52181.0 | 61621.5 | 59799.5 | **65836.5** |
| MONTEZUMA'S REVENGE | 0.0 | **4753.3** | 0.0 | 0.0 | 0.0 | 0.0 | 0.0 | 384.0 | 0.0 | 2643.5 | 2643.5 |
| MS. PAC-MAN | 307.3 | **6951.6** | 3085.6 | 2711.4 | 6283.5 | 6518.7 | 3327.3 | 5380.4 | 4416.9 | 2724.3 | 3749.2 |
| NAME THIS GAME | 2292.3 | 8049.0 | 8207.8 | 10616.0 | 11971.1 | 12270.5 | **15572.5** | 13136.0 | 12636.5 | 9907.2 | 9543.8 |
| PHOENIX | 761.4 | 7242.6 | 8485.2 | 12252.5 | 23092.2 | 18992.7 | 70324.3 | **108528.6** | 10261.4 | 40092.2 | 46536.4 |
| PITFALL! | -229.4 | **6463.7** | -286.1 | -29.9 | 0.0 | -356.5 | 0.0 | 0.0 | -3.7 | -3.5 | -8.9 |
| PONG | -20.7 | 14.6 | 19.5 | 20.9 | **21.0** | 20.6 | 20.9 | 20.9 | 20.7 | 20.7 | 20.6 |
| PRIVATE EYE | 24.9 | **69571.3** | 146.7 | 129.7 | 103.0 | 200.0 | 206.0 | 4234.0 | 15198.0 | 15177.1 | 15188.8 |
| Q*BERT | 163.9 | 13455.0 | 13117.3 | 15088.5 | 19220.3 | 16256.5 | 18760.3 | **33817.5** | 21222.5 | 22956.5 | 21509.2 |
| RIVER RAID | 1338.5 | 17118.0 | 7377.6 | 14884.5 | 21162.6 | 14522.3 | 20607.6 | **22920.8** | 16957.3 | 16608.3 | 17380.7 |
| ROAD RUNNER | 11.5 | 7845.0 | 39544.0 | 44127.0 | 69524.0 | 57608.0 | 62151.0 | 62041.0 | 66790.5 | 71168.0 | **111310.0** |
| ROBOTANK | 2.2 | 11.9 | 63.9 | 65.1 | 65.3 | 62.6 | 27.5 | 61.4 | **71.8** | 68.5 | 70.4 |
| SEAQUEST | 68.4 | 42054.7 | 5860.6 | 16452.7 | **50254.2** | 26357.8 | 931.6 | 15898.9 | 5071.6 | 8425.8 | 20994.1 |
| SKIING | -17098.1 | **-4336.9** | -13062.3 | -9021.8 | -8857.4 | -9996.9 | -19949.9 | -12957.8 | -10632.9 | -10753.4 | -10870.6 |
| SOLARIS | 1236.3 | **12326.7** | 3482.8 | 3067.8 | 2250.8 | 4309.0 | 133.4 | 3560.3 | 2236.0 | 2760.0 | 2099.6 |
| SPACE INVADERS | 148.0 | 1668.7 | 1692.3 | 2525.5 | 6427.3 | 2865.8 | 15311.5 | 18789.0 | 2387.1 | 2448.6 | 10153.9 |
| STARGUNNER | 664.0 | 10250.0 | 54282.0 | 60142.0 | 89238.0 | 63302.0 | 125117.0 | 127029.0 | 48942.0 | 70038.0 | 79521.5 |
| SURROUND | -10.0 | 6.5 | -5.6 | -2.9 | 4.4 | 8.9 | 1.2 | **9.7** | 0.9 | 6.7 | 7.0 |
| TENNIS | -23.8 | -8.3 | 12.2 | -22.8 | 5.1 | 0.0 | 0.0 | 0.0 | 23.4 | 23.3 | **23.6** |
| TIME PILOT | 3568.0 | 5229.2 | 4870.0 | 8339.0 | 11666.0 | 9197.0 | 7553.0 | 12926.0 | 18871.5 | **19401.0** | 18841.5 |
| TUTANKHAM | 11.4 | 167.6 | 68.1 | 218.4 | 211.4 | 204.6 | 245.9 | 241.0 | 263.2 | 272.6 | **275.4** |
| UP'N DOWN | 533.4 | 11693.2 | 9989.9 | 22972.2 | 44939.6 | 16154.1 | 33879.1 | 125754.6 | **194989.5** | 64354.2 | 70790.4 |
| VENTURE | 0.0 | 1187.5 | 163.0 | 98.0 | 497.0 | 54.0 | 48.0 | 5.5 | 0.0 | 1597.5 | **1653.5** |
| VIDEO PINBALL | 16256.9 | 17667.9 | 196760.4 | 309941.9 | 98209.5 | 282007.3 | 479197.0 | 533936.5 | 261720.2 | 469366.0 | 496101.0 |
| WIZARD OF WOR | 563.5 | 4756.5 | 2704.0 | 7492.0 | 7855.0 | 4802.0 | 12352.0 | 17862.5 | 18484.0 | 13170.5 | **19530.5** |
| YARS' REVENGE | 3092.9 | 54576.9 | 18098.9 | 11712.6 | 49622.1 | 11357.0 | 69618.1 | 102557.0 | 109607.5 | 102760.0 | **148855.0** |
| ZAXXON | 32.5 | 9173.3 | 5363.0 | 10163.0 | 12944.0 | 10469.0 | 13886.0 | 22209.5 | 16525.0 | 25215.5 | **27582.5** |

