# OpenReview forum: "The Reactor: A fast and sample-efficient Actor-Critic agent for  Reinforcement Learning"
_ICLR.cc/2018/Conference — Accept (Poster)_

### Official Review · AnonReviewer3 · 2017-11-27
**A promising "Rainbow-like" combination of deep RL techniques, unfortunately with no proper comparison to Rainbow**

**Rating:** 7
**Confidence:** 4

**Review:**

This paper proposes a novel reinforcement learning algorithm (« The Reactor ») based on the combination of several improvements to DQN: a distributional version of Retrace, a policy gradient update rule called beta-LOO aiming at variance reduction, a variant of prioritized experience replay for sequences, and a parallel training architecture. Experiments on Atari games show a significant improvement over prioritized dueling networks in particular, and competitive performance compared to Rainbow, at a fraction of the training time.

There are definitely several interesting and meaningful contributions in this submission, and I like the motivations behind them. They are not groundbreaking (essentially extending existing techniques) but are still very relevant to current RL research.

Unfortunately I also see it as a step back in terms of comparison to other algorithms. The recent Rainbow paper finally established a long overdue clear benchmark on Atari. We have seen with the « Deep Reinforcement Learning that Matters » paper how important (and difficult) it is to properly compare algorithms on deep RL problems. I assume that this submission was mostly written before Rainbow came out, and that comparisons to Rainbow were hastily added just before the ICLR deadline: this would explain why they are quite limited, but in my opinion it remains a major issue, which is the main reason why I am advocating for rejection.

More precisely, focusing on the comparison to Rainbow which is the main competitor here, my concerns are the following:
- There is almost no discussion on the differences between Reactor and Rainbow (actually the paper lacks a « related work » section). In particular Rainbow also uses a version of distributional multi-step, which as far as I can tell may not be as well motivated (from a mathematical point of view) as the one in this submission (since it does not correct for the « off-policyness » of the replay data), but still seems to work well on Atari.
- Rainbow is not distributed. This was a deliberate choice by its authors to focus on algorithmic comparisons. However, it seems to me that it could benefit from a parallel training scheme like Reactor’s. I believe a comparison between Reactor and Rainbow needs to either have them both parallelized or none of them (especially for a comparison on time efficiency like in Fig. 2)
- Rainbow uses the traditional feedforward DQN architecture while Reactor uses a recurrent network. It is not clear to which extent this has an impact on the results.
- Rainbow was stopped at 200M steps, at which point it seems to be overall superior to Reactor at 200M steps. The results as presented here emphasize the superiority of Reactor at 500M steps, but a proper comparison would require Rainbow results at 500M steps as well.

In addition, although I found most of the paper to be clear enough, some parts were confusing to me, in particular:
- « multi-step distributional Bellman operator » in 3.2: not clear exactly what the target distribution is. If I understand correctly this is the same as the Rainbow extension, but this link is not mentioned.
- 3.4.1 (network architecture): a simple diagram in the appendix would make it much easier to understand (Table 3 is still hard to read because it is not clear which layers are connected together)
- 3.3 (prioritized sequence replay): again a visual illustration of the partitioning scheme would in my opinion help clarify the approach

A few minor points to conclude:
- In eq. 6, 7 and the rest of this section, A does not depend (directly) on theta so it should probably be removed to avoid confusion. Note also that using the letter A may not be best since A is used to denote an action in 3.1.
- In 3.1: « Let us assume that for the chosen action A we have access to an estimate R(A) of Qπ(A) » => « unbiased estimate »
- In last equation of p.5 it is not clear what q_i^n is
- There is a lambda missing on p.6 in the equation showing that alphas are non-negative on average, just before the min
- In the equation above eq. 12 there is a sum over « i=1 »
- That same equation ends with some h_z_i that are not defined
- In Fig. 2 (left) for Reactor we see one worker using large batches and another one using many threads. This is confusing.
- 3.3 mentions sequences of length 32 but 3.4 says length 33.
- 3.3 says tree operations are in O(n ln(n)) but it should be O(ln(n))
- At very end of 3.3 it is not clear what « total variation » is.
- In 3.4 please specify the frequency at which the learner thread downloads shared parameters and uploads updates
- Caption of Fig. 3 talks about « changing the number of workers » for the left plot while it is in the right plot
- The explanation on what the variants of Reactor (ND and 500M) mean comes after results are shown in Fig. 2.
- Section 4 starts with Fig. 3 without explaining what the task is, how performance is measured, etc. It also claims that Distributional Retrace helps while this is not the case in Fig. 3 (I realize it is explained afterwards, but it is confusing when reading the sentence « We can also see... »). Finally it says priorization is the most important component while the beta-LOO ablation seems to perform just the same.
- Footnote 3 should say it is 200M observations except for Reactor 500M
- End of 4.1: « The algorithms that we compare Reactor against are » => missing ACER, A3C and Rainbow
- There are two references for « Sample efficient actor-critic with experience replay »
- I do not see the added benefit of the Elo computation. It seems to convey essentially the same information as average rank.

And a few typos:
- Just above 2.1.3: « increasing » => increasingly
- In 3.1: « where V is a baseline that depend » => depends
- p.7: « hight » => high, and « to all other sequences » => of all other sequences
- Double parentheses in Bellemare citation at beginning of section 4
- Several typos in appendix (too many to list)

Note: I did not have time to carefully read Appendix 6.3 (contextual priority tree)

Edit after revision: bumped score from 5 to 7 because (1) authors did many improvements to the paper, and (2) their explanations shed light on some of my concerns

---

> ### Author Response · Authors · 2017-12-15
> **Authors' response to AnonReviewer3**
>
>  Thank you very much for your helpful review.
>
> >> There is almost no discussion on the differences between Reactor and Rainbow
> >> I assume that this submission was mostly written before Rainbow came out, and that comparisons to Rainbow were hastily added just before the ICLR deadline
>
> Admittedly, the comparisons with Rainbow were less detailed than we would have liked. Please note that Rainbow was put on Arxiv only three weeks before the ICLR submission deadline. However we have already included experimental comparisons with Rainbow, both in the form of presenting the learning curves and final evaluations. We will add a more in-depth comparison with Rainbow and discussion of related work in the appendix.
>
> >> I believe a comparison between Reactor and Rainbow needs to either have them both parallelized or none of them.
>
> Rainbow works on GPUs, Reactor works on CPUs. A single GPU is not equivalent to a single CPU. Parallelizing Rainbow is out of the scope of this work. First, because this was not the focus of our work. Second, because it would be a non-trivial task potentially worth publication on its own. More generally, the same parallelization argument would also apply to comparisons between A3C and DQN.
>
> >> Rainbow uses the traditional feedforward DQN architecture while Reactor uses a recurrent network. It is not clear to which extent this has an impact on the results.
>
>
> There are many differences between Rainbow and Reactor: 1) LSTM vs frame stacking, 2) actor-critic vs value-based algorithm 3) beta-LOO vs Q-learning, 4) Retrace vs n-step learning, 5) sequence prioritization vs transition prioritization, 6) entropy bonus vs noisy networks. Reactor is not an incremental improvement of Rainbow and is a completely different algorithm. This makes it impractical to compare on a component-by-component basis. For the most important contributions we performed an ablation study within Reactor’s framework, but naturally we can not ablate every architectural choice that we have made.
>
> >> Rainbow was stopped at 200M steps, at which point it seems to be overall superior to Reactor at 200M steps.
>
> This is not correct. In the human-starts evaluation Reactor significantly outperforms Rainbow at 200M steps. In the no-op-starts evaluation Rainbow significantly outperforms Reactor at 200M steps. Both Reactor and Rainbow were trained with 30 random no-op-starts. Their evaluation with 30 random human starts shows how well each algorithm generalizes to new initial conditions. We would argue that the issues of generalization here are similar to those seen between training and testing error in supervised learning. We thus show that Reactor generalizes better to these unseen starting states.
>
> >> (network architecture): a simple diagram in the appendix would make it much easier to understand
>
> We will add the diagram to the supplementary material.
>
> >> again a visual illustration of the partitioning scheme would in my opinion help clarify the approach
>
> We will add an illustration to the supplementary material. We will also correct all other typos mentioned in the review. Thank you for taking note of them.

---

> > ### Comment · AnonReviewer3 · 2018-01-10
> > **Final thoughts**
> >
> > Thanks!
> >
> > I can definitely imagine it was hard to make a proper comparison to Rainbow within such a short timeframe. I still think such a comparison would be quite valuable, to better evaluate the impact of their respective unique components. I'm afraid we are back to a situation where it's not clear what works best -- I guess that's the curse of the Atari benchmark.
> >
> > I appreciate the many improvements to the paper (though I lack time to look at them thoroughly), in particular the Appendix section on the comparisons with Rainbow. I admit I had read your paper as a DQN extension, while it makes more sense to see it as an A3C extension. I'll change my score to acceptance.
> >
> > NB: I disagree with the statement that "In the human-starts evaluation Reactor significantly outperforms Rainbow at 200M steps". It has slightly higher median normalized score, but lower Elo score. I don't think we can draw a solid conclusion from this (like claiming that "Reactor generalizes better to these unseen starting states").
> >
> > Also if you can fix this typo in a final version, it looks like you added a "i=1" in eq. 12's sum, but forgot its upper bound.

---

### Official Review · AnonReviewer1 · 2017-11-27
**interesting work with several contributions and large experiments with some but not all recent approaches**

**Rating:** 7
**Confidence:** 2

**Review:**

This paper proposes a novel reinforcement learning algorithm containing several contributions made by the authors: 1) a policy gradient algorithm that uses value function estimates to improve the policy gradient, 2) a distributed multi-step off-policy algorithm to estimate the value function, 3) an experience replay buffer mechanism that can handle sequences and (4) a distributed architecture, where threads are dedicated to either learning or interracting with the environment. Most contributions consist in improvements to handle multi-step trajectories instead of single step transitions. The resulting algorithm is evaluated on the ATARI domain and shown to outperform other similar algorithms, both in terms of score and training time. Ablation studies are also performed to study the interest of the 4 contributions.

I find the paper interesting. It is also well written and reasonably clear. The experiments are large, although I was disappointed that PPO was not included in the evaluation, as this algorithm also trains much faster than other algorithms.

quality
+ several contributions
+ impressive experiments

clarity
- I found the replay buffer not as clear as the other parts of the paper.
. run time comparison: source of the code for the baseline methods?
+ ablation study showing the merits of the different contributions
- Methods not clearly labeled. For example, what is the difference between Reactor and Reactor 500M?

originality
+ 4 contributions

significance
+ important problem, very active area of research
+ comparison to very recent algorithms
- but no PPO in the evaluation

---

> ### Author Response · Authors · 2017-12-15
> **Authors response to AnonReviewer1**
>
> Thank you very much for your review and recognising novelty of our contributions.
>
> >> I found the replay buffer not as clear as the other parts of the paper.
>
> We will do our best to clarify the description, most likely in the appendix given space limitations.
>
> >> Methods not clearly labeled. For example, what is the difference between Reactor and Reactor 500M?
>
> We will clarify the labels. `Reactor 500M` denotes the performance of Reactor at 500 million training steps.
>
> >> but no PPO in the evaluation
>
> The PPO paper did not present results at 200M frames but at 40M frames, and their results seem to be weaker than ACER on 40M frames: ACER was better than PPO on 28/49 games tested. For the purpose of comparison to other algorithms, we chose to evaluate all algorithms at (at least) 200M frames, and Reactor is much better than ACER on 200M frames. Unfortunately, we don’t know how PPO perform at 200M frames, so a direct comparison is impossible.

---

### Official Review · AnonReviewer2 · 2017-11-27
**Nice integration of recent deep RL advances with decent empirical results**

**Rating:** 7
**Confidence:** 4

**Review:**

This paper presents a new reinforcement learning architecture called Reactor by combining various improvements in
deep reinforcement learning algorithms and architectures into a single model. The main contributions of the paper
are to achieve a better bias-variance trade-off in policy gradient updates, multi-step off-policy updates with
distributional RL, and prioritized experience replay for transition sequences. The different modules are integrated
well and the empirical results are very promising. The experiments (though limited to Atari) are well carried out and
the evaluation is performed on both sample efficiency and training time.

Pros:
1. Nice integration of several recent improvements in deep RL, along with a few novel tricks to improve training.
2. The empirical results on 57 Atari games are impressive, in terms of final scores as well as real-time training speed.

Cons:
1. Reactor is still less sample-efficient than Rainbow, with significantly lower scores after 200M frames. While the
reactor trains much faster, it does use more parallel compute, so the comparison with Rainbow on wall clock time is
 not entirely fair. Would a distributed version of Rainbow perform better in this respect?
2. Empirical comparisons are restricted to the Atari domain. The conclusions of the paper will be much stronger if
results are also shown on other environments like Mujoco/Vizdoom/Deepmind Lab.
3. Since the paper introduces a few new ideas like prioritized sequence replay, it would help if a more detailed analysis
 was performed on the impact of these individual schemes, even if in a model simpler than the Reactor. For instance, one could investigate the impact of prioritized sequence replay in models like multi-step DQN or recurrent DQN. This will help us understand  the impact of each of these ideas in a more comprehensive fashion.

---

> ### Author Response · Authors · 2017-12-15
> **Author response to AnonReviewer2**
>
> We were happy to see that the reviewer recognised the novelty both the introduced ideas (prioritization, distributional Retrace and the beta-LOO policy gradient algorithm) and integration of the ideas into a single agent architecture.
>
> >> Reactor is still less sample-efficient than Rainbow, with significantly lower scores after 200M frames
>
> This is not correct. In the human-starts evaluation Reactor significantly outperforms Rainbow at 200M steps. In the no-op-starts evaluation Rainbow significantly outperforms Reactor at 200M steps. Both Reactor and Rainbow were trained with 30 random no-op-starts. Their evaluation with 30 random human starts shows how well each algorithm generalizes to new initial conditions. We would argue that the issues of generalization here are similar to those seen between training and testing error in supervised learning. We thus show that Reactor generalizes better to these unseen starting states.
>
> >> While the Reactor trains much faster, it does use more parallel compute, so the comparison with Rainbow on wall clock time is not entirely fair.
>
> The reviewer is right in the sense that Reactor executes more floating point operations per second, but it trains much shorter in wall time resulting in an overall similar number of computations executed. We make no claim that Reactor uses overall less computational operations to train an agent. Nevertheless, we believe that having a fast algorithm in terms of wall time is important because of the potential to shorten experimentation time. The measure is still informative, as one may choose Reactor over Rainbow when multiple CPU machines are available (as opposed to a single GPU machine).
>
> >>  Empirical comparisons are restricted to the Atari domain.
>
> We focused on Atari domain to facilitate the comparison to the prior work.
>
> >> Since the paper introduces a few new ideas like prioritized sequence replay, it would help if a more detailed analysis was performed on the impact of these individual schemes
>
> The paper already contains the ablation study comparing relative importances of individual components. Since the number of novel contributions is large (beta-LOO, distributional retrace, prioritized sequence replay), it is difficult to explore all possible configurations of the components.

---

> > ### Comment · AnonReviewer2 · 2018-01-12
> > **Final thoughts**
> >
> > Thanks to the authors for their response. As I mentioned in the initial review, I think the method is definitely promising and provides improvements. My comments were more on claims like "Reactor significantly outperforms Rainbow" which is not evident from the results in the paper (a point also noted by Reviewer 3). These claims could be made more specific, with appropriate caveats, or additional experiments could be performed to help substantiate the claims better.

---

### Author Response · Authors · 2018-01-05
**New revision.**

We have just added a new revision addressing the reviewer comments, which we much appreciate.

---

### Decision · Program_Chairs · 2018-01-29
**ICLR 2018 Conference Acceptance Decision**

**Decision:**

Accept (Poster)

**Comment:**

This paper presents a nice set of results on a new RL algorithm. The main downside is the limitation to the Atari domain, but otherwise the ablation studies are nice and the results are strong.